

# Improved pointing information for SCIAMACHY from in-flight measurements of the viewing directions towards sun and moon

Klaus Bramstedt[1], Thomas C. Stone[2], Manfred Gottwald[3], Stefan Noël[1], Heinrich Bovensmann[1], and John P. Burrows[1]

[1]Institute of Environmental Physics (IUP), University of Bremen, Otto-Hahn-Allee 1, 28359 Bremen, Germany
[2]US Geological Survey, 2255 North Gemini Drive, Flagstaff, AZ 86001, U.S.A
[3]German Aerospace Center, Remote Sensing Technology Institute, Münchner Str. 20, 82234 Wessling, Germany

*Correspondence to:* K. Bramstedt (klaus.bramstedt@uni-bremen.de)

**Abstract.**

The SCanning Imaging Absorption spectroMeter for Atmospheric CHartographY (SCIAMACHY) on Envisat (2002-2012) performed nadir, limb, solar/lunar occultation and various monitoring measurements. The pointing information of the instrument is determined by the attitude information of the Envisat platform with its star trackers together with the encoder readouts of both the azimuth and the elevation scanner of SCIAMACHY.

In this work, we present additional sources of attitude information from the SCIAMACHY measurements itself. The basic principle is the same as used by the star tracker: We measure the viewing direction towards celestial objects, i.e. sun and moon to detect possible mispointings.

In sun over limb port observations, we utilise the vertical scans over the solar disk. In horizontal direction, SCIAMACHY's sun follower device (SFD) is used to adjust the viewing direction. Moon over limb port measurements use for both the vertical and the horizontal direction the adjustment by the SFD. The viewing direction is steered towards the intensity centroid of the illuminated part of the lunar disk. We use reference images from the USGS Robotic Lunar Observatory (ROLO) to take into account the inhomogeneous surface and the variations by lunar libration and phase to parameterise the location of the intensity centroid from the observation geometry. Solar observations through SCIAMACHY's so-called sub-solar port (with a viewing direction closely to zenith) use in vertical direction also the SFD. In horizontal direction the geometry of the port defines the viewing direction.

Using these three type of measurements, we fit improved mispointing parameters by minimising the pointing offsets in elevation and azimuth. The geolocation of all retrieved products will benefit from this, especially the altitudes assigned to SCIAMACHY's limb and occultation products will be improved.

# 1 Introduction

Satellite measurements allow global observations of atmospheric constituents. Passive remote sensing instruments use three basic types of geometry: nadir, limb and occultation. Critical point in the latter two is always the pointing knowledge, i.e. the precise knowledge of the viewing direction and with that the altitude grid of the observations.



SCIAMACHY (SCanning Imaging Absorption spectroMeter for Atmospheric CHartographY) is a passive remote sensing moderate-resolution imaging UV-Vis-NIR spectrometer on board the European Space Agency's (ESA) Environmental Satellite (Envisat). Envisat was launched in March 2002 from Kourou, French Guiana to a sun synchronous orbit, where it accomplished 10 years successful mission. In April 2012, the mission ended due to a platform malfunction. SCIAMACHY observes Earth's atmosphere in nadir, limb and solar/lunar occultation geometries and provides column and profile information of atmospheric parameters of relevance to ozone chemistry, air pollution, and climate monitoring issues (Burrows et al., 1995; Bovensmann et al., 1999; Gottwald and Bovensmann, 2011).

SCIAMACHY's solar occultation measurements are done in the northern hemisphere, when the sun rises due to the orbital motion of the platform. On the day side of the orbit, alternating limb and nadir measurements are performed. The same airmasses are probed first in limb and a few minutes later in nadir. Lunar occultation is made in the southern hemisphere, if the waxing moon with a phase larger than 0.5 is in the field of view and the moon rise occurs after sunset. Dark current measurements are performed during eclipse. Various types of monitoring measurements are executed in the daily, weekly and monthly calibration periods.

The pointing information for the instrument is derived from the attitude information of the platform, which itself is calibrated using the three star trackers of Envisat. Here, we use SCIAMACHY's own measurements to further improve the pointing knowledge. In Bramstedt et al. (2012), precise pointing knowledge has been derived for SCIAMACHY's solar occultation measurements. It used the sun over limb port measurements, which are the solar occultation measurements continued above the atmosphere. In this work, we will additionally use the moon over limb port measurements and the sun over sub-solar port observations to derive further pointing information. The viewing direction of the instrument towards the sun or moon is derived from these measurements. From the attitude information of the platform, the viewing direction towards sun or moon is calculated. Systematic differences between the measured and the calculated viewing directions indicate a mispointing. Appropriate mispointing parameters are needed, which improve the attitude information of the platform and the SCIAMACHY line of sight information.

The first set of in-flight mispointing parameters has been derived in 2007 by Gottwald et al. (2007), used in SCIAMACHY data products (re-)processed since 2010. Priority was minimising the observed large tangent height offsets of about 1 km observed in the limb measurements. The sun over limb port and over sub-solar port measurements were used for this. In Bramstedt et al. (2012), a remaining systematic offset in tangent height of 280 m, a horizontal offset of 5 km, and seasonal pattern in both offsets have been shown for solar occultation data.

In this work, the combination of the pointing measurements from all three measurement types listed above is used for fitting new mispointing parameters, which further improve the pointing accuracy for all SCIAMACHY measurements.

Section 2.1 introduces the quantities and the parts of the instrument relevant for this work. The relevant measurement types are described in Section 3. The variability of the lunar disk needs special care as described Sect. 4. Finally, all measurements are utilised to derive improved mispointing parameters in Sect. 5. The appendices adds technical details of the algorithm.



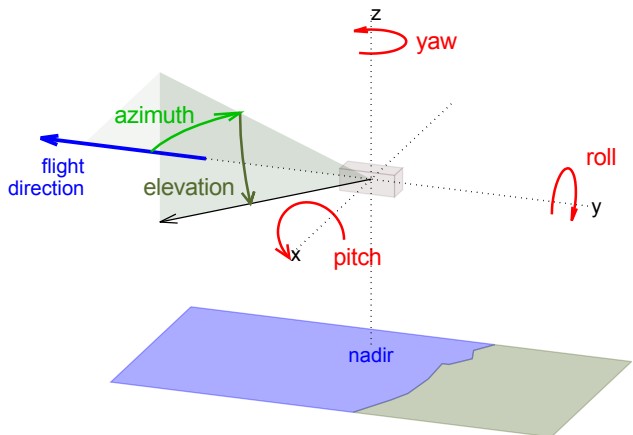

**Figure 1.** Sketch defining the attitude angles (roll, pitch, yaw) and the viewing angles (azimuth, elevation).

## 2 The instrument

### 2.1 Coordinate Systems

Envisat's orbit and viewing geometry can be described in various coordinate systems (CS), as defined in Alvarez (1997). The primary platform fixed CS is Envisat's Satellite Relative Reference CS. A possible mis-alignment between platform and instrument is described by a set of three mispointing angles. They define the consecutive rotations in roll, pitch, and yaw to transform Satellite Relative Reference CS to the Satellite Relative Actual Reference (SRAR) CS. Initial mispointing angles for the SCIAMACHY instrument were already determined in the on-ground calibration.

The instrument's viewing direction is expressed as azimuth and elevation angle in the SRAR coordinate system. Figure 1 shows a sketch of these quantities.

### 2.2 The scanner unit

The scan mirror system comprises the elevation scan mechanism (ESM) and the azimuth scan mechanism (ASM). Both scan mechanisms have on one side an aluminium mirror and on the other side a diffuser for the nominal solar measurements. Figure 2 is a schematic sketch of the orientation of the scanners. The ESM rotates parallel to the the roll axis of the instrument, whereas the ASM rotates parallel to the yaw axis. For the nominal nadir measurements, only the ESM is used. For limb and occultation measurements, both mirrors are needed, as shown in Fig. 2. The position of ESM and ASM (if used) defines the actual viewing direction of the instrument.

### 2.3 The detectors

The spectrometer has eight detector modules with 1024 detector pixels each, covering the ultraviolet, visible and near-infrared spectral range. Seven broadband detectors, the Polarisation Measurement Devices (PMDs), are used to determine the polarisa-



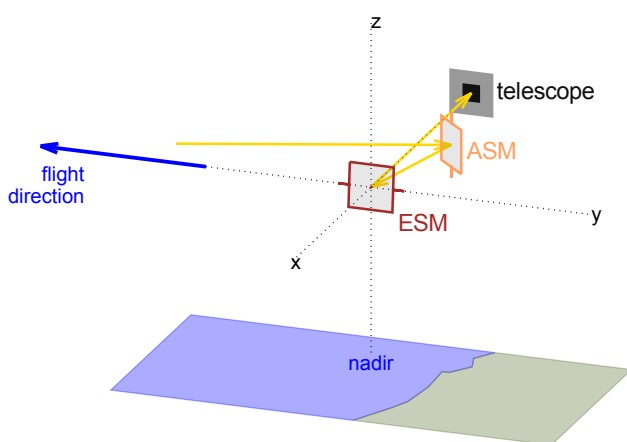

**Figure 2.** Schematic sketch of SCIAMACHY's scanner unit. The yellow beam follows the limb light path.

tion state of the incoming light. PMD 1-6 measure the light polarised perpendicular to SCIAMACHY's optical plane. PMD 7 is sensitive to the 45° component in the visible spectral range. The PMDs are sampling devices, whereas the spectral detectors are integrating devices. The PMDs provide with 40 Hz the highest readout frequency. Here, we use only PMD 4 (800–900 nm) to measure the intensity, as in Bramstedt et al. (2012).

### 2.4 The viewing ports

The limb port is opened in flight direction towards Earth's horizon. It is used for SCIAMACHY's limb and occultation measurements. Both scan mirrors are used to adjust the viewing direction in elevation and azimuth.

The nadir port allows a downward viewing direction for SCIAMACHY's nadir measurement. Only the ESM mirror is used for this viewing direction.

The so-called sub-solar port is directed upwards, about 29° off the zenith. The sun is observed through this port to monitor
the nadir light path, therefore only the ESM is used in the sub-solar measurements. The azimuth of the viewing direction is fixed at 270° by the opening of the port.

### 2.5 SCIAMACHY's sun follower device

The sun follower device (SFD) can be used to steer the instrument's line of sight towards the central part of the intensity distribution of the sun or the moon, further on named intensity centroid (IC). The SFD is a four quadrant sensor and detects
the sun or the moon in a 2.2° × 2.2° wide field. The image of the sun or moon is reflected from the polished blades of the spectrometer entrance slit onto the SFD. The SFD detector has a broadband sensitivity centred around 500 nm.



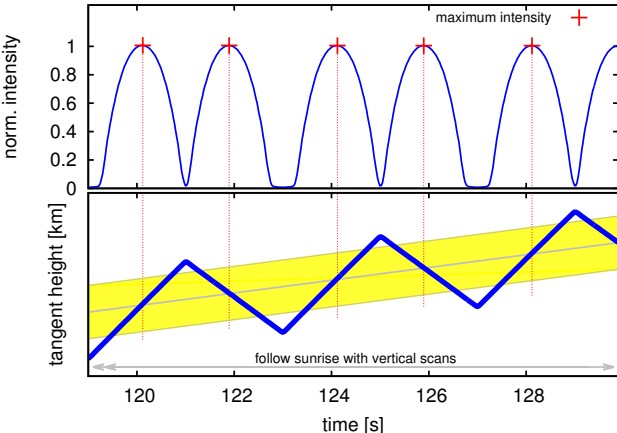

**Figure 3.** Sun over limb port measurement sequence. Bottom: In elevation, the solar disk (yellow area) is scanned by movements of the ESM mirror (blue line), which overall follows the rising sun. In total, 36 scans are usable for the pointing investigations. Top: By fitting the maximum intensity in each scan, the elevation angle of the centre of sun is measured for the time of maximum intensity.

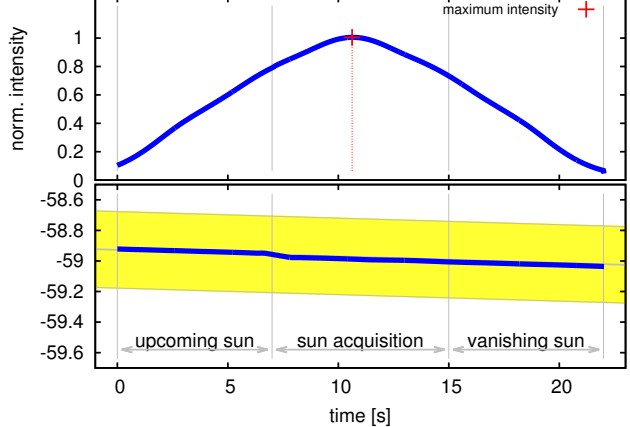

**Figure 4.** Sun over sub-solar port measurement sequence. Bottom: After an initial phase of waiting for the sun, the SFD tracks the sun in elevation. Top: The ASM mirror is not in the light path. The azimuth angle of the solar disk is changed by the orbital motion of the platform. At maximum intensity, the azimuth angle is 270° because that is the viewing direction of the sub-solar port in azimuth.

## 3   Measurements

Three types of measurements are used to derive additional pointing information: Sun and Moon observed through the limb port (the solar/lunar occultation measurements at atmospheric tangent heights) and the sun observed through the sub-solar port. Due to observing geometry defined by the orbit and the viewing ports, sun over limb port is performed above northern hemisphere, moon over limb port above the southern hemisphere and the sun over sub-solar port above the equatorial region.



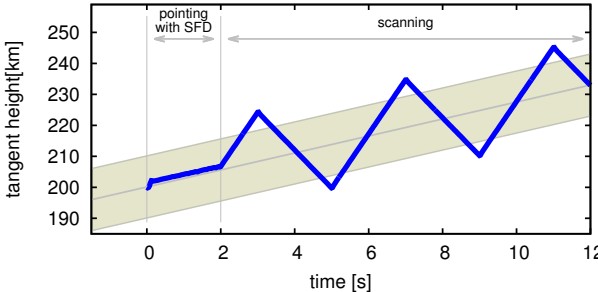

**Figure 5.** Moon over limb port measurement sequence. The moon is tracked for two seconds with the SFD in both elevation and azimuth direction. The following scans over the lunar disk are not used here.

The sun over limb port measurement (Fig. 3) follows the rising sun, performing vertical scans over the solar disk. The viewing direction towards the centre of the sun in elevation is determined by fitting the maximum intensity during the scan. In azimuth, the sun is tracked with the SFD. Details are given in Bramstedt et al. (2012). Sun over over limb port is the upper part of the solar occultation measurement performed every orbit. Once per day, a shortened solar occultation measurement is performed, which ends with a two seconds period with tracking by the SFD also in elevation. The difference between the two methods is below 1 mdeg (Bramstedt et al., 2012).

Three different sequences for sun over sub-solar port measurement are done. In the sequence used here, the sun is tracked in elevation by the SFD (Fig. 4). The ASM is not involved in this light path. The sun moves in azimuth direction through the field of view because of the orbital motion of the satellite. At the maximum intensity during the measurement, the sun is at $270\,^{\circ}$ azimuth angle (the opening direction of the sub-solar port). This sun over sub-solar port measurement is part of the monthly calibration. After October 2006, it was scheduled every three days.

The moon over limb port measurement (Fig. 5) used here is a monitoring measurement, which is performed well above the atmosphere. In its first phase, the moon is tracked in azimuth and elevation by the SFD. This phase is used here. The SFD adjusts the viewing direction towards the intensity centroid (IC) of the actual lunar disk. The location of the IC is discussed in Sect. 4. This monitoring measurement is performed once per day, if the moon appears in the field of view. This is the case for about one week per month during the phase of the waxing moon with a lunar phase larger than 0.5. Note: the regular lunar occultation measurements are performed every orbit with the same restrictions. Additionally, the tangent point has to be at local night time, limiting the usable period to about 6 months per year.

## 4 Lunar variability

### 4.1 ROLO reference images

The Robotic Lunar Observatory (ROLO) of U.S. Geological Survey (USGS) is located in Flagstaff, Arizona in the United States. In 6 years of operation (1997–2003), the observatory has acquired over 110.000 lunar images. Twin telescopes cover





the visible and near-infrared range (350–950 nm) in 23 bands and the short-wave infrared (950–2350 nm) in 9 bands (Kieffer

and Stone, 2005). We use the calibrated images of the 550 nm band, which is the closest band to the sensitivity of the SFD at

500 nm. Size of the images is 512×512 pixel.

We simulate the control loop of the SFD to determine the IC for the lunar images. The image is separated in four quadrants

(representing the sectors of the SFD). The boundaries between the left and right and the upper and lower part of the images are

adjusted until the difference between the left and right part on the one hand and the upper and lower part on the other and is

65 minimised. After this adjustment, the sums over the intensities of the pixels in each quadrant are equal within the limits of the

discretisation. The determined IC is independent from rotations of the image. After a 90° rotation, the geometry is repeated

because of the sensor symmetry. Taking into account the mirror symmetry of the sensor at the 45° axis, we have to check only

rotations between 0 and 45°. We rotated a high resolution lunar image in 5° steps from 0 to 45° and determined in all cases

the same IC location.

We determined the IC for 15 ROLO images close to full moon. Figure 6 shows the images with the boundaries of the

quadrants after adjustment. The cross of the boundaries marks the intensity centre of the moon, where the SFD will adjust the

line of sight.

From the SFD simulation, we get the location of the IC as pixel position on the ROLO image. The corresponding lunar

coordinates are derived by visual inspection of a high resolution lunar image. We used the program *Virtual Moon Atlas Pro*

*V6.0* (Legrand and Chevalley, 2013a, b) for this purpose. Figure 7 shows an example: the green dot marks the identified location

in the Ptolemaeus crater region.

## 4.2 Parametrisation of the lunar intensity centroid

The observing geometry is defined by the sub-observer point $P_{\mathrm{sobs}}$, the intersection of the straight line from the observer (here

Envisat or the ROLO telescope) to the centre of the moon with the lunar surface. This point defines the visible part of the lunar

surface, its change with time is called the libration of the moon.

The lunar illumination is defined by the sub-solar point $P_{\mathrm{ssol}}$, the intersection of the straight line from the centre of the sun to

the centre of the moon with the lunar surface. The distance between the sub-solar point and the sub-observer point determines

the illumination of the visible part of the lunar surface: the lunar phase.

For the 15 ROLO images, we have calculated $P_{\mathrm{sobs}}$, $P_{\mathrm{ssol}}$ (using the HORIZONS software, see Appendix A2) and deter-

5 mined $P_{\mathrm{icen}}$. The lunar coordinates in latitude and longitude of these points are given in Table 1.

In Figure 8 these points are plotted in a Cartesian coordinate system. We neglected the in principal spherical nature of the

problem, because the deviation to the Cartesian approach is small for regions near the equator. The directions and distances

between $P_{\mathrm{sobs}}$ and $P_{\mathrm{ssol}}$ are similar in all cases and varies depending on the location of $P_{\mathrm{ssol}}$. With the following equation,

the location of the $P_{\mathrm{icen}}$ can be parametrised:

$$P_{\mathrm{icen}} = P_{\mathrm{sobs}} + d + a(P_{\mathrm{ssol}} - P_{\mathrm{sobs}}) \qquad (1)$$



**Table 1.** The ROLO reference images with sub-observer and sub-solar point coordinates, and the lunar phase as illumination in percent. The intensity centroid *IC image* is determined from the SFD simulation. The *IC parameter* is calculated with the fitted parameters. The distance in degree between the two ICs is given in the last two columns. The distance standard deviation in latitude is $0.51°$, in longitude $0.42°$.

| image | measurement time | sub-observer | | sub-solar | | illum. | IC image | | IC param | | IC distance | |
| --- | --- | --- | --- | --- | --- | --- | --- | --- | --- | --- | --- | --- |
| | | lon | lat | lon | lat | [%] | lon | lat | lon | lat | lon | lat |
| mm194311 | 10-May-1998 08:46:21 | -2.04 | -4.89 | 11.90 | -1.51 | 98.448 | 5.70 | -10.04 | 5.87 | -9.58 | -0.16 | -0.45 |
| mm194403 | 11-May-1998 05:49:54 | -2.41 | -5.59 | 1.21 | -1.52 | 99.775 | 1.44 | -10.04 | 1.55 | -10.18 | -0.11 | 0.14 |
| mm194407 | 11-May-1998 07:52:51 | -2.87 | -5.51 | 0.17 | -1.52 | 99.809 | 1.06 | -10.33 | 0.87 | -10.11 | 0.20 | -0.22 |
| mm194411 | 11-May-1998 09:51:13 | -3.29 | -5.48 | -0.83 | -1.52 | 99.834 | 0.78 | -10.35 | 0.22 | -10.09 | 0.56 | -0.26 |
| mm197401 | 10-Jun-1998 05:54:53 | -4.11 | -5.64 | -5.28 | -1.53 | 99.861 | -2.36 | -10.76 | -1.98 | -10.23 | -0.38 | -0.53 |
| mm197403 | 10-Jun-1998 06:52:47 | -4.31 | -5.60 | -5.77 | -1.53 | 99.858 | -2.49 | -10.76 | -2.29 | -10.19 | -0.20 | -0.57 |
| mm212109 | 04-Nov-1998 06:42:59 | 0.65 | 6.74 | -0.15 | 1.47 | 99.784 | 3.48 | 0.14 | 2.92 | 0.69 | 0.55 | -0.55 |
| mm215006 | 03-Dec-1998 11:26:07 | 1.51 | 6.69 | 4.58 | 1.47 | 99.721 | 6.06 | 1.11 | 5.26 | 0.65 | 0.80 | 0.46 |
| mm218002 | 02-Jan-1999 09:45:45 | 4.21 | 4.42 | 0.67 | 1.02 | 99.817 | 5.43 | -1.93 | 5.43 | -1.33 | 0.00 | -0.60 |
| mm226801 | 31-Mar-1999 08:55:54 | 3.51 | -3.96 | 10.20 | -1.24 | 99.604 | 8.29 | -8.24 | 8.64 | -8.76 | -0.35 | 0.52 |
| mm244610 | 25-Sep-1999 08:45:19 | -5.15 | 5.45 | -3.41 | 1.19 | 99.839 | -1.96 | 0.14 | -1.91 | -0.44 | -0.06 | 0.58 |
| mm250509 | 23-Nov-1999 06:23:57 | -1.08 | 6.39 | -0.85 | 1.40 | 99.810 | 1.70 | 0.26 | 1.58 | 0.39 | 0.12 | -0.13 |
| mm262206 | 19-Mar-2000 06:24:17 | 4.31 | -4.37 | 15.88 | -1.33 | 98.917 | 11.65 | -8.79 | 11.31 | -9.12 | 0.35 | 0.33 |
| mm303606 | 07-May-2001 07:12:22 | 5.07 | -4.72 | 8.30 | -1.21 | 99.827 | 8.23 | -9.07 | 8.88 | -9.39 | -0.65 | 0.32 |
| mm303611 | 07-May-2001 09:53:02 | 4.62 | -4.51 | 6.94 | -1.21 | 99.876 | 7.44 | -8.24 | 8.08 | -9.21 | -0.65 | 0.97 |

**Table 2.** The fitted parameters $d$ and $a$ to determine the location of the lunar intensity centroid from the observing geometry with Eq. (1).

$$d_{lon} = 2.575 \pm 0.151 \qquad a_{lon} = 0.382 \pm 0.028$$
$$d_{lat} = -5.223 \pm 0.129 \qquad a_{lat} = 0.156 \pm 0.032$$

$d$ is the distance between $P_{\mathbf{sobs}}$ and $P_{\mathbf{ssol}}$ for full moon, $a$ parametrises the movement of the IC due to the lunar phase. All terms in Eq. (1) are two element vectors with the lunar coordinates in longitude and latitude.

$d$ and $a$ are fitted by minimising the distances between the parametrised and the observed ICs. The fitted parameters are given in Table 2. In Figure 8, the crosses show the parameterised ICs. With the parametrisation of Eq. (1), we can calculate the IC for all lunar observation close to full moon from the observing geometry only.

The uncertainty of the IC location depends on three components. First, the ROLO reference images have a resolution of $512 \times 512$ pixels. Each image pixel has an instantaneous field of view of 1.13 mdeg (measured precisely from astrometry of star fields). This pixel width is the uncertainty for the location of the IC. Second, the parameterised IC matches not exactly the IC of the image. The standard deviation of the distances between these two ICs in lunar longitude / latitude is $0.43°$/ $0.51°$. As standard deviation in viewing angle from the satellite, this is at maximum 2.3 mdeg. Third, we have the pointing precision of



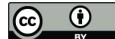

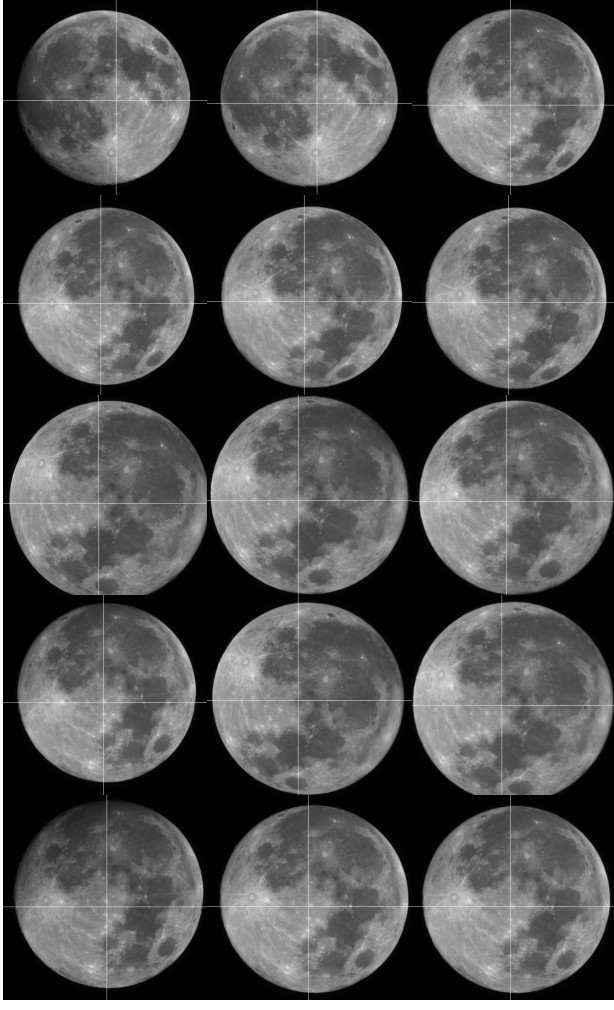

**Figure 6.** The 15 ROLO reference images. The white lines mark the boundaries of the four quadrants in the SFD simulation after adjustment. The images are listed in Table 1 from top left to bottom right.

the SFD of 1 mdeg as estimated in Bramstedt et al. (2012). Adding up the three error sources as root of the summed squares of the error, we roughly estimate the uncertainty in the viewing angle to 2.8 mdeg. In elevation, this corresponds to a tangent height uncertainty of about 160 m.

25  ## 5  Mispointing analysis

Mispointing of SCIAMACHY can have two main causes. First, the assumed mispointing angles to determine the SRAR coordinate system may not be correct, either because the SCIAMACHY mis-alignment is not well described or because the platform attitude knowledge is inaccurate. The SRAR CS can be aligned by appropriate roll, pitch, and yaw mispointing angles.





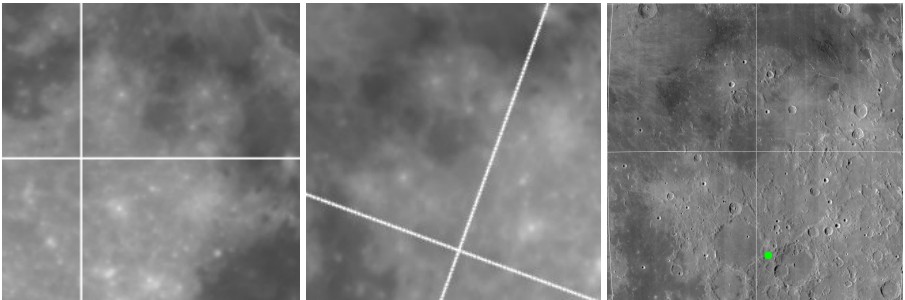

**Figure 7.** An example of the determination of the IC lunar coordinates with the program *Virtual Moon*. The left panel is a clipping of the ROLO image *mm194407* including the adjusted boundaries of the four quadrants in the SFD simulation. The middle panel is the left panel rotated by about 70 ° counter-clockwise to match the orientation of the moon in *Virtual Moon*. The right panel shows the same area as image in *Virtual Moon*. The intensity centre in the Ptolemaeus crater region is marked by the green dot. The shown region in the middle and left panel corresponds to the one used in Fig. 8.

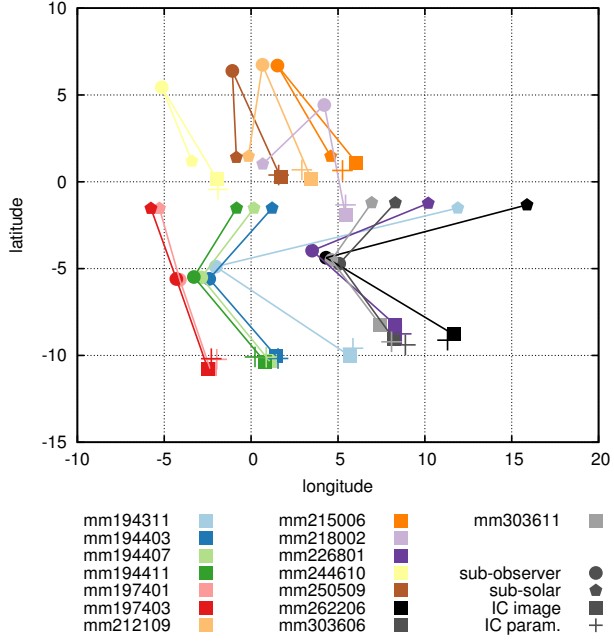

**Figure 8.** Illustration of the IC parametrisation. For each ROLO reference image, the sub-observer point $P_{\mathrm{sobs}}$, the sub-solar point $P_{\mathrm{ssol}}$ and the IC location in the ROLO images $P_{\mathrm{icen}}$ are plotted onto a grid of lunar coordinates. The parameterised IC (shown as cross) is in all cases close to the observed intensity centre.

Second, the scan mirror positions might have an offset. As the ASM rotates in the yaw axis, an ASM offset corresponds to a
30  yaw mispointing, if the ASM mirror is used in the measurement. An ESM offset correspond to a roll mispointing. However, if also the ASM is used (limb light path), an ESM offset corresponds to a mispointing in pitch, see also Figures 1 and 2.





**Table 3.** Envisat and SCIAMACHY mispointing parameters. The columns are: **AOCS**: These mispointing parameters are part of Attitude and Orbit Control System (AOCS) of Envisat and describe the initial platform mispointing, here only shown for comparison. **scanner control**: SCIAMACHY mispointing as determined before launch. These mispointing angles are also used in the scanner control of SCIAMACHY. **extra mispointing**: In the analysis Gottwald et al. (2007), the mispointing was derived relative to the mispointing already implemented in the scanner control. **scia mispointing**: These four columns describe the mispointing parameters as used in the L0-1 processing. **launch**: For the first processor versions since launch, no mispointing was assumed. **1st 2007**: First attempt to recognise the 2007 analysis. **final 2007**: final settings from the 2007 analysis, which should be the sum of *scanner control* and *extra mispointing*. **new fit**: Mispointing parameters as determined by this work. Beyond mispointing angles in pitch, roll and yaw, also offsets to the scanner positions ($dpos_{ESM}$, $dpos_{ASM}$) are determined. Version 9 is the proposed processor version to implement the new mispointing parameters.

| | *AOCS* | *scanner control* | *extra mis-pointing* | launch | 1st 2007 | final 2007 | **new fit** |
|---|---|---|---|---|---|---|---|
| | | | | | —— SCIAMACHY mispointing —— | | |
| L0-1b processor version | *all* | *all* | – | <6.0 | ≥6.03 | ≥7.0 | ≥9.0 |
| pitch [mdeg] | *-167.074* | *0.630* | *-26±3* | 0.0 | -26.0 | -25.4 | **-17.06 ±0.63** |
| roll [mdeg] | *50.233* | *1.662* | *-20±1* | 0.0 | -20.0 | -18.3 | **-25.18 ±0.48** |
| yaw [mdeg] | *391.987* | *-227.464* | *9±8* | 0.0 | -218.46 | -218.5 | **-227.61 ±1.26** |
| $dpos_{ESM}$[mdeg] | | | | | | | **-8.11 ±0.54** |
| $dpos_{ASM}$[mdeg] | | | | | | | **-102.02 ±1.43** |

In summary, we have five parameters to minimise the mispointing: the three mispointing angles and two offsets for the mirror positions.

Sub-solar measurements are available once per month, therefore we selected roundabout one measurement per type and month during the mission. Excluded is the year 2002, because the first sub-solar measurements failed due to a timing problem. In total 96 moon over limb port, 96 sun over limb port and 90 sun over sub-solar measurements are used.

A non-linear Levenberg-Marquad fit is used to minimise elevation and azimuth angle offsets by varying the five mispointing parameters. Starting points are the currently used final values from 2007. The newly fitted mispointing parameters are listed in Table 3. For completeness, also the previously used mispointing parameters are given as explained in the table caption.

Fig. 9 shows the improvement. For the measurements used in the fit, the observed elevation / azimuth angle offset (dλ / dφ) is plotted, once using the mispointing parameters from 2007 and once using the new parameters (marked as 2016). The elevation and azimuth angle offsets for all three measurement types are clearly reduced for the full time series. In elevation, the offsets with opposite signs for the lunar and solar measurements over limb port have been corrected. In azimuth, the large horizontal offset for the measurements over the limb light path has been removed.

The new parameters briefly have the following impact on the pointing:

– The ASM mirror offset fixes the large azimuth angle offset.





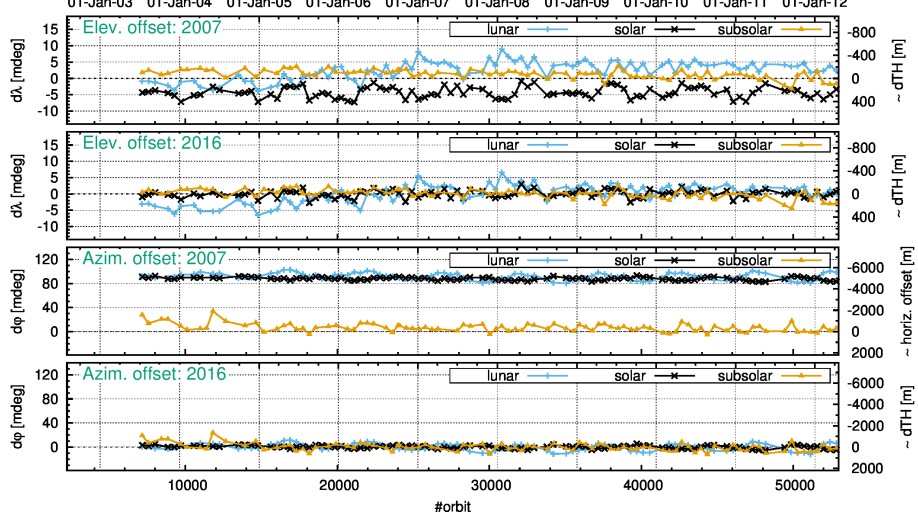

**Figure 9.** The elevation / azimuth angle offsets (dλ / dφ) for the measurements used in the fit of the new mispointing parameters. Top and second panel: The elevation angle offset using the mispointing parameters from 2007 and the newly fitted parameters (2016). Third and fourth panel show the azimuth angle offset for both cases. With the new parameters, the offsets are strongly reduced.

- The yaw mispointing is almost unchanged because the sub-solar azimuth angle had no offset.

- The changed roll mispointing angle fixes the elevation offsets with opposite signs for solar and lunar occultation, because these two measurement have different viewing directions in azimuth. Whereas the solar occultation is performed at $\sim -30°$ azimuth angle, the lunar occultation azimuth angle is up to $\sim +40°$.

- The pitch mispointing angle results in an overall adjustment of the elevation angle offset.

## 5.1 Impact of new parameters

In Bramstedt et al. (2012), the mispointing for the solar occultation measurements has been investigated in detail. With the new mispointing parameters, we repeated this analysis. Figure 10 shows the viewing angles offsets for the previous and the new parameters. The mean tangent height / horizontal offset has been reduced from 276 m / 5031 m to 29 m / 71 m. The encoder readouts of the scanner unit are discretized in mdeg steps, corresponding to about 56 m at the tangent point. The remaining offsets are at the noise level of these readouts. The amplitude of the seasonal variation of the elevation angle offset has been reduced from 122 m to 48 m. Because of the roll mispointing, the elevation angle offset was azimuth dependent. Therefore, the fixed seasonal pattern of the azimuth of the solar occultation measurement contributed strongly to the seasonal pattern in the elevation offsets. The offsets and seasonal variations are summarised in Table 4.



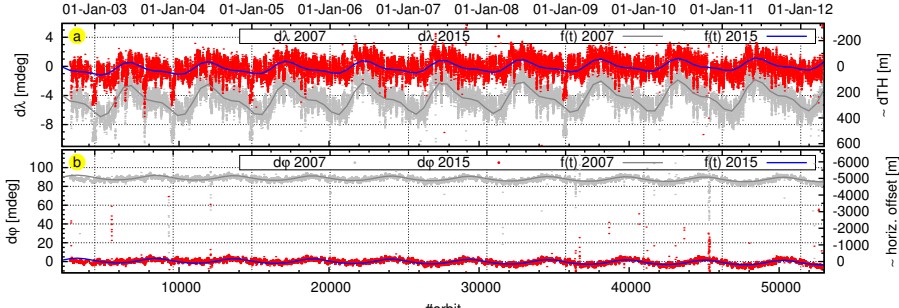

**Figure 10.** Solar occultation mispointing with mispointing parameters as defined in 2007 (grey) and with the newly fitted mispointing parameters (labelled 2016). a) Elevation angle offset $d\lambda$. b) Azimuth angle offset $d\varphi$. The lines are a function with a constant, a linear, and a annual and semiannual sinusoidal term, fitted to the datasets (details see Bramstedt et al. (2012)).

**Table 4.** The offset and amplitudes of the seasonal cycles of the functions fitted to the occultation mispointing angle with the mispointing parameters as defined in 2007 (grey) and with the newly fitted mispointing parameters (labelled 2016), see also Fig. 10. The values for the 2007 mispointing very slightly differ from the ones given in Bramstedt et al. (2012) because of the update in the orbit parameters (see Sect. A1) and a slightly changed set of available orbits in the level 1b product version 8.

|  | amplitude | | mean offset | |
|---|---|---|---|---|
|  | [mdeg] | [m] | [mdeg] | [m] |
| Elev 2007 | 2.158±0.011 | 122±1 | -4.89±0.013 | 276±1 |
| Elev 2016 | 0.849±0.012 | 48±1 | -0.51±0.013 | 29±1 |
| Azim 2007 | 2.230±0.017 | 126±1 | 89.04±0.019 | -5031±1 |
| Azim 2016 | 2.257±0.017 | 128±1 | 1.25±0.019 | -71±1 |

# 6  Conclusions

We provide a method to determine improved mispointing parameters for SCIAMACHY on-board ENVISAT from in-flight measurements. The viewing direction towards sun or moon is derived for three types of measurements and compared to the expected viewing direction. These types are sun and moon over limb port and sun over sub-solar port measurements.

SCIAMACHY's sun follower device adjusts the azimuth direction in sun over limb port, the elevation direction in sun over sub-solar port and both directions in moon over limb port observations. Sun over limb port elevation offset are derived from

the scans over the solar disk: at maximum intensity the instrument points to the solar centre. Similarly, the solar disk moves in azimuth over the field of view of the sub-solar port by the orbital motion, again the maximum intensity defines the direction towards the solar centre.

The moon is an inhomogeneous and variable target, and the SFD points to the intensity centroid (IC) of the moon instead of the lunar centre. We investigated reference images from the USGS Robotic Lunar Observatory (ROLO) and derived a

parameterisation for the location of the IC from the observation geometry only.





The viewing directions are compared to the ones according to the attitude information of the platform to derive azimuth and elevation angle offsets. Mispointing angle in pitch, roll and yaw for the platform can correct the assumed attitude. Zero offsets for the positions of the scan mirrors can improve the pointing of the instrument.

With a suitable subset of the available pointing measurements, we minimise the mispointing for the whole mission by varying

the five mispointing parameters in a non-linear Levenberg-Marquad fit.

The azimuth and elevation angle offsets observed with the previous set of mispointing parameters are clearly reduced by the newly derived set. For sun over limb port measurements, the mean offset in tangent height is reduced from 276 m to negligible 29 m. The amplitude of the seasonal cycle of the tangent height offset is reduced from 122 m to 48 m. The previously observed horizontal offset of 5 km is reduced to 71 m.

The new mispointing angles and the mirror position offsets according Table 3 will be used in the next release of the SCIA-MACHY data processor and improve the geolocation for all measurements. Most important is the better knowledge of the tangent heights for SCIAMACHY's limb and occultation measurements.

## Appendix A: Software packages

### A1  Envisat-1 mission CFI software

All calculations necessary for orbit propagation and pointing issues for Envisat are combined in the *Envisat-1 mission CFI software* (Muñoz, 2011), named CFIs further on. The software needs as input the orbit information of the satellite. In Bramstedt et al. (2012), the orbit state vector from the main product header in the SCIAMACHY level 1b product has been propagated along the orbit. In this work, we use the DORIS precise orbit product files for Envisat, which are more accurate and also used for the re-processing campaigns of SCIAMACHY data. The DORIS system (Jayles et al., 2006) provides accurate orbit

information, for Envisat the radial orbit precision is 2 cm (Willis et al., 2006).

### A2  HORIZONS

*The JPL HORIZONS on-line solar system data and ephemeris computation service* of the Solar System Dynamics (SSD) group of NASA's Jet Propulsion Laboratory (JPL) (Giorgini et al., 1996; Yeomans et al., 1996) provides access to key solar system data and flexible production of highly accurate ephemerides for solar system objects. It is needed for the specific requirements

of the lunar target, which are not covered by the CFIs. Specifically, the system can calculate the sub-solar and sub-observer point on the moon for each viewing point in space and time. It can also calculate observation parameters for target points on the moon specified by lunar coordinates.

## Appendix B: Detailed algorithm for lunar measurements

The CFI software does not provide all quantities necessary for determining the viewing direction toward the IC on the moon.

The sequence is therefore as follows:



    – The orbital positions of the satellite during the lunar measurement is calculated using the CFIs.

    – The sub-observer point from this satellite position and the sub-solar point in lunar coordinates are derived with the HORIZONS system.

    – Using the parametrisation of Eq. (1), the lunar coordinates of the IC is derived.

– The HORIZONS system calculates for the coordinates the right ascension and the declination of the IC for the satellite position.

    – These astronomical coordinates are used in the CFIs to derive the actual azimuth and elevation angle of the IC.

*Acknowledgements.* SCIAMACHY is a national contribution to the ESA Envisat project, funded by Germany, The Netherlands, and Belgium. SCIAMACHY data have been provided by ESA. This work has been funded by DLR Space Agency (Germany), by ESA via the SCIAMACHY Quality Working Group (SQWG) and by the state and the University of Bremen.





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
