# Peer review of "Improved pointing information for SCIAMACHY from in-flight measurements of the viewing directions towards sun and moon"

_Atmospheric Measurement Techniques, 2016_

## Referee Comment (RC1) · Anonymous Referee #1 · 9 Feb 2017

The manuscript describes a new method to assess the mispointing of SCIAMACHY on Envisat and proposes an improved correction scheme for future calibration. The paper is well written, detailed enough for non-experts to follow, but avoiding unnecessary details. The method explained is convincing, and the results seem reasonable. I would like to congratulate the authors. It was a pleasure to read this paper.

I have only a few very minor comments, some of them even on the pure technical side. They are listed below. I recommend the paper for rapid publication after correction of these minor issues.

Specific/technical comments:

[Figure]

One remark concerning all the manuscript: The authors make quite extensive use of abbreviations like CS, ESM, ASM, PMD, SFD, IC etc. As far as I have checked, all these abbreviations are properly explained at their first use. Nevertheless, it would be easier for the reader if somewhat less extensive use of abbreviations would be made. This is just a suggestion.

Abstract: At the end of the abstract, some quantitative information on the amount of mispointing identified and the correction to be applied should be added; for the non-expert reader, it would also be helpful to translate these numbers into the tangent height correction and the remaining tangent height uncertainty.

Page 2, line 48/49: "observed" appears twice in this sentence, and one of them should be removed.

Page 6, line 39: remove one "over"

Page 11, line 38: "Starting points": do you mean "initial guess"?

Page 13, Fig. 10: there is only a label "2015" in the figure, but in the caption a label "2016" is mentioned. One of both seems to need correction.

Page 12/13, section 5.1 and Conclusions: For the non-expert reader it would be of interest how the new corrections translate into tangent height corrections, and what the remaining tangent height uncertainties are. It would be very valuable to add this information.

---

## Referee Comment (RC2) · Anonymous Referee #3 · 3 May 2017

This paper is aimed at study of pointing information for SCIMACHY on board ENVISAT. The paper is well written and can be published as it stands. The authors may modify caption to Fig.10.
* * *

---

## Author Comment (AC1) · 29 May 2017

The authors thank the referees for there encouraging reviews. We adress the remaining points in detail below. Citations from the referees comments are in *italic font*. New text in the paper is presented in blue.

**Answers to referee #1:**

*One remark concerning all the manuscript: The authors make quite extensive use of abbreviations like CS, ESM, ASM, PMD, SFD, IC etc. As far as I have checked, all these abbreviations are properly explained at their first use. Nevertheless, it would be easier for the reader if somewhat less extensive use of abbreviations would be made.*

[Figure]

*This is just a suggestion.*

We removed the abbrevation CS and replaced it by the full term 'coordinate system', which indeed is not necessary here as an abbreviation. We think, the other technical abbrevations should not be replaced by their full terms, otherwise the text would be much longer. Instead, we added a small table with the remaining abbreviations for the readers convenience:

**Table 1.** List of abbreviations used in this paper.

| | |
|---|---|
| SCIAMACHY | SCanning Imaging Absorption spectroMeter for Atmospheric CHartographY (on-board Envisat). |
| ESM | SCIAMACHY's Elevation scan mechanism. |
| ASM | SCIAMACHY's Azimuth scan mechanism. |
| SFD | SCIAMACHY's Sun follower device. |
| PMD | SCIAMACHY's Polarisation measurement device. |
| ROLO | The Robotic Lunar Observatory of U.S. Geological Survey (USGS). |
| IC | Intensity centroid (of the actual lunar disk). |
| SRAR | Satellite Relative Actual Reference (Envisat coordinate system). |

*Abstract: At the end of the abstract, some quantitative information on the amount of mispointing identified and the correction to be applied should be added; for the non-expert reader, it would also be helpful to translate these numbers into the tangent height correction and the remaining tangent height uncertainty.*

See answer further below.

*Page 2, line 48/49: "observed" appears twice in this sentence, and one of them should be removed.*

Done.

*Page 6, line 39: remove one "over"*

Done.

*Page 11, line 38: "Starting points": do you mean "initial guess"?*

Yes, "initial guess" is the correct term. We changed it in the text.

*Page 13, Fig. 10: there is only a label "2015" in the figure, but in the caption a label "2016" is mentioned. One of both seems to need correction.*

'2015' in the Figure was wrong. To be consistent with Figure 9 and the text, this is changed to 2016.

*Page 12/13, section 5.1 and Conclusions: For the non-expert reader it would be of interest how the new corrections translate into tangent height corrections, and what the remaining tangent height uncertainties are. It would be very valuable to add this information.*

We added a new figure, which shows the difference in tangent height and elevation for one typical orbit with limb, lunar and solar occulation measurements. The figure is added in section 5.1. We added appropriate text parts in this section, the abstract and the conclusions:

**New Figure:**
See Fig. 1 below.

**Section 5.1:**
The impact on the limb and occultation tangent heights is shown for orbit 15193 (25 January 2005) in Fig. 11. Given is the difference in tangent heights (elevation angles) between the configuration derived in this paper and the previous one. The solar occultation tangent heights are changed by about $-290\,\mathrm{m}$ ($+5\,\mathrm{mdeg}$), for lunar occultation we observe about $+140\,\mathrm{m}$ ($-2.5\,\mathrm{mdeg}$). The SCIAMACHY limb measurements have a horizontal swath of about $960\,\mathrm{km}$, for a large part of the spectral range divided into

four vertical profiles. Depending on the profile (and therefore the azimuth angle), the changes are in the range of -50 m to +60 m ($+1$ to $-1.2$ mdeg). An east-west asymmetry in the tangent height offset has been observed in former tangent height retrievals from the limb measurements. The limb tangent height retrieval method is described in Kaiser et al. (2004) and von Savigny et al. (2005). The corrections derived in this paper confirm an east-west asymmetry. Depending on season, the solar occultation tangent heights changes are in the range of -330 to -130 m. The range for solar occultation is 0 to +130 m. Limb tangent heights are changed in the range of -60 to +70 m, with a fixed pattern in the horizontal direction. For all tangent points, the horizontal location is moved by about 5 km.

**Abstract:**

The altitudes assigned to SCIAMACHY's solar occultation measurements are changed by -130 − -330 m, lunar occultation are changed by 0 − +130 m, and the limb measurements are changed by -50 − +60 m (depending on season, altitude, and azimuth angle). The horizontal location of the tangent point is for all measurement changed by about 5 km. These updates are implemented in version 9 of the SCIAMACHY Level 1b products and Level 2 version 7 (based in L1b version 9).

**Conclusions:**

The tangent height altitudes of SCIAMACHY's solar occultation measurements are changed by -130 − -330 m, lunar occultation is altered by 0 − +130 m. The limb measurements are changed by -50 − +60 m, including the removal of an east-west asymmetry in the limb tangent height offsets. The horizontal location of the tangent point is for all three cases changed by about 5 km.

**Referee #2:**

*The authors may modify caption to Fig.10.*

The year 2015 used in the figure as label was wrong. To be consistent with Figure 9 and the text, this is label changed to 2016. To clarify the message of this plot, we added also this sentence to the figure caption:

With the newly fitted parameters, offsets for both angles are close to zero.

[Figure]

**Fig. 1.** Tangent height changes for the limb and occultation measurements of orbit 15193 (25-Jan-2005) from the previous mispointing parameters to the ones derived here.